# Molecular Dynamics Simulations of the Mechanical Properties of Cellulose Nanocrystals—Graphene Layered Nanocomposites

**DOI:** 10.3390/nano12234170

**Published:** 2022-11-24

**Authors:** Xingli Zhang, Zhiyue Chen, Liyan Lu, Jiankai Wang

**Affiliations:** 1College of Mechanical and Electrical Engineering, Northeast Forestry University, Harbin 150040, China; 2College of Underwater Acoustic Engineering, Harbin Engineering University, Harbin 150009, China

**Keywords:** cellulose nanocrystals, graphene, layered nanocomposites, molecular dynamics simulation, mechanical properties

## Abstract

Cellulose nanocrystals (CNCs) have received a significant amount of attention due to their excellent physiochemical properties. Herein, based on bioinspired layered materials with excellent mechanical properties, a CNCs-graphene layered structure with covalent linkages (C-C bond) is constructed. The mechanical properties are systematically studied by molecular dynamics (MD) simulations in terms of the effects of temperature, strain rate and the covalent bond content. Compared to pristine CNCs, the mechanical performance of the CNCs-graphene layered structure has significantly improved. The elastic modulus of the layered structure decreases with the increase of temperature and increases with the increase of strain rate and covalent bond coverage. The results show that the covalent bonding and van der Waals force interactions at the interfaces play an important role in the interfacial adhesion and load transfer capacity of composite materials. These findings can be useful in further modeling of other graphene-based polymers at the atomic scale, which will be critical for their potential applications as functional materials.

## 1. Introduction

Cellulose nanocrystals (CNCs) show exceptional mechanical properties, biodegradability and environmental friendliness [1,2,3]. The typical CNCs with lengths of 50–4000 nm and widths of 3–20 nm are extracted using sulfuric acid hydrolysis from natural cellulose microfibrils (e.g., trees, plants, cotton, agricultural waste) [4,5,6,7]. CNCs have been used in the development of network composites and as reinforced fillers in polymer composites [8,9,10,11]. The upcoming applications make it essential to gain a fundamental understanding of the mechanical properties of CNCs. The experimentally measured tensile strengths and elastic modulus of CNC nanomaterials were reported to be 2–50 GPa and 105–220 GPa, respectively [12,13]. The wide distribution in the reported experimental values stems primarily from differences in uncontrolled measurement conditions and material sources. Molecular dynamics (MD) simulation has also been successfully used to predict the structure and mechanics of CNCs at molecular scale [14,15,16,17]. The simulation can capture the deformation and microstructure evolution of the materials and the stress response of the tensile strain by physical statistical methods. Many MD simulation results show that the mechanical properties of CNCs are associated with the size of the simulation system, strain rate and temperature [18,19,20].

CNCs with a high degree of crystallinity can easily be formed into smooth films with uniform thickness by appropriate treatment [21,22,23]. The CNC films have been extensively used in biodegradable electronics, batteries, and optical devices [24,25,26,27,28]. However, the high crystallinity of CNCs also makes them prone to deformation and fracture under the action of external forces. For this reason, a CNCs-graphene layered structure is introduced to further improve the mechanical strength of CNCs. The source of inspiration for this nanocomposite is the nacre-like layered composites with outstanding mechanical properties. The extraordinary properties of natural nacre are attributed to the synergistic strengthening effects of the interface interactions [29,30]. Typical interface categories of layered materials can be divided into hydrogen bonds, ionic bonds, π-π interactions, and covalent bonding. Generally, the interface strength formed by covalent bonding is higher than other bonds [31,32,33,34]. The study of the mechanical properties of layered materials is becoming increasingly popular in academia. Park et al. developed an approach for combining graphene oxide (GO) with polyallylamine (PAA) by covalent bonding, resulting in a 30% increase in Young’s modulus and a 10% increase in tensile strength [35]. Tian et al. experimentally studied the improved mechanical capabilities of polydopamine (PDA)/GO nanosheet composites. The results indicated that high strength covalent bonding occurred between the GO nanosheets and PDA polymers [36]. The above studies reveal that graphene-based layered materials with stronger interfaces can be produced and applied in the reinforcement of mechanical characteristics.

In this study, the mechanical properties of CNCs-graphene layered nanocomposites with covalent bonding are investigated utilizing non-equilibrium molecular dynamics (NEMD) under typical temperature and strain rate. The effects of covalent bonding on the interfacial interactions are also predicted by comparing the elastic modulus of CNCs-graphene layered structure with that of pristine CNCs. In addition, the mechanism behind the mechanical property is further explained with the help of the atomic configuration. The simulation results could provide some important connections between mechanical behavior and covalent bonding in layered nanocomposites, and it could provide some insights for optimizing the mechanical properties of other polymers.

## 2. Models and Simulation Method

The NEMD simulations were carried out using the LAMMPS (Large-scale Atomic/Molecular Massively Parallel Simulator) program package [37]. All interatomic interactions were described by ReaxFF potential function which was expected to accurately reproduce the mechanical properties of a wide range of systems [38]. The intramolecular potential energies were determined using bond orders calculated from bonded interactions and non-bonded interactions (such as van der Waals and coulombic interactions). The parameters used in ReaxFF were obtained by optimizing a training set of quantum-mechanics calculation data [29]. The simulation size of the CNCs-graphene layered structure was 6.4 nm(X) × 2.3 nm(Y) × 3.8 nm(Z). The periodic boundary condition was applied in all directions. As shown in Figure 1, the covalent linkages were distributed between graphene and CNCs layers with different coverages varying from 0% to 8%. They were generated by randomly removing H atoms in CNC chains and then the bonds were bonded to a carbon atom of the graphene layer.

The NEMD simulations were conducted by first minimizing the energy of the system using the conjugate gradient method. The time step was set to 0.5 fs for all the simulation processes. The simulation temperature was controlled by using a Nosé-Hoover thermostat. Before starting the tensile simulation, the nanostructures reached the global energy minimum using the steepest descent minimization algorithm. After equilibration, the system was stretched for 200 ps under the NVT ensemble. Energy minimization was performed after each deformation increment to maintain the lowest energy state of the system. For each initial configuration, the axial tensile loading modeling was repeated three times with different velocity initialization seeds. The elastic modulus calculated according to Hooke’s law was obtained by averaging the results of three independent simulation trials.

## 3. Results and Discussion

### 3.1. Effect of Temperature on Mechanical Property

The calculated stress–strain behaviors of the pristine CNCs and CNCs-graphene layered structure subject to x-direction strain are shown in Figure 2. The curves are obtained under a tensile strain rate of 3 × 10^−4^ fs^−1^ at temperatures of 300 K, 400 K, and 500 K. As shown in Figure 2a,b, the fracture strain for loading on pristine CNCs and CNCs-graphene structure are 5.1% and 7.4%, respectively. The simulation result of pristine CNCs in this paper are in agreement with the previously reported results of crystalline cellulose obtained using uniform deformation and nanoscale indentation methods [21]. This indicates that the simulation models constructed in this paper are valid.

Figure 3 shows the effect of temperature on the elastic modulus for pristine CNCs and CNCs-graphene layered structure. It can be seen that the elastic modulus of all models experiences a primary decrease with increasing temperature. The elastic modulus of the CNCs-graphene layered structure is approximately three times higher than that of pristine CNCs. This phenomenon can be attributed to the covalent bonding interactions and van der Waals forces between graphene and CNC layers, leading the nanocomposite to have a higher resistance to external forces. The impact of temperature on the elastic modulus can be related to an increase in the system kinetic energy with the increasing temperature, which weaken the van der Waals forces and the covalent interactions [23,28].

In order to further verify the above results, the atomic configurations of two nanocomposites under tensile stresses with the increasing temperature are shown in Figure 4. For the pristine CNC structure, plastic deformation occurs gradually with increasing temperature. At 500 K, the disordered region of pristine CNCs expands to the whole simulation zone and the structure is nearly separated into two parts. This tensile stress of pristine CNCs may be mainly derived from van der Waals and hydrogen bonding interactions. However, van der Waals forces at the level of the cellulose associations are not strong enough to endure the strain. Hydrogen bonds and intra-chain hydrogen bonds in pristine CNCs are also destroyed due to the increasing temperature, which makes the CNC chains more easily deformed [39]. The CNCs-graphene layered structure has better structural stability than that of the pristine CNCs with increasing temperature. For the CNCs-graphene layered structure, the arrangement of atoms in the graphene layer appears to be distorted. The covalent bonds between CNCs and graphene layers are completely broken, but the CNC chains are observed to have no apparent breakage. This phenomenon could be explained by the greater robustness of covalent bonding compared with linear molecules. When the temperature is increased to 500 K, the broken covalent bonding results in high tensile strength along with curling of the graphene layer.

### 3.2. Effect of Strain Rate on Mechanical Property

The stress-strain behavior of the two nanocomposites with different strain rates at the temperature of 300 K is shown in Figure 5. It is observed that their fracture strain values all increase with the increasing strain rates, and the values for CNCs-graphene layered composite are much higher than those of pristine CNCs. A similar trend was reported by Wu et al. [18] who studied the impact of strain rate on Iβ crystalline cellulose obtained using MD simulation. Vlassiouk et al. [40] prepared macroscale polymer/graphene laminates using an experimental method, and the fracture strain was around 5%, which was also consistent with our simulation results. The ReaxFF force field used in this paper could not match the nanocomposites accurately; all the values of the liner region are considered to calculate the elastic modulus for eliminating the singularity phenomenon in that local region as much as possible [41]. In addition, some negative stress is observed in the two nanocomposites during the tensile process which usually appears under compression conditions. This is likely due to the local compression in the structures, which should be induced by the cellulose chain breaking.

The elastic modulus of pristine CNCs and CNCs-graphene layered composites are given in Figure 6. The strain rates have an inconspicuous effect on the elastic modulus of pristine CNCs; however, the elastic modulus of CNCs-graphene layered composite increases with the increasing strain rates. This result indicates that CNCs-graphene layered composite can endure a high strain rate during the tensile process. The improved mechanical properties of layered composites under high tensile strain rate are also ascribed to their more stable structure because of the covalent bonds.

To understand the dependence of strain rate on the mechanical properties of the two nanocomposites, the configuration of atoms with the increasing strain rates are shown in Figure 7. The atomic structure does not change significantly when the strain rates are small, although the lattice spacing increases. When the strain rate is equal to 3 × 10^–4^/fs, an obvious local stress concentration appears in the pristine CNCs. Some local fractures of graphene layer in layered composite also appear. This fact proves that the covalent bonds at the interface can strengthen the stress transfer capacity. As the displacement in the tensile direction increases, an attractive restoring force arising from the covalent bonds in CNCs-graphene layered composite attempts to hold the layers together. Moreover, van der Waals forces also provide some interface strengths due to the large specific surface area between CNCs and graphene layers [42].

### 3.3. Effect of Covalent Bond Coverage on Mechanical Property

The mechanical properties of the CNCs-graphene layered structure at each content level of covalent bonds are shown in Figure 8 and Figure 9. When the bond content increases from 0% to 8%, the fracture strain increases from 7.2% to 10.5%, and the measured elastic modulus increases from 414.79 GPa to 448.2 GPa. These results also prove that the interfacial adhesion and load transfer effect of composite materials are significantly enhanced through covalent bonds. It may also be indicated that the covalent bonds between CNCs and graphene layers have greater strength than the polymer matrix [43].

## 4. Conclusions

In this work, the mechanical properties of the CNCs-graphene layered structure with covalent bonding are investigated by molecular dynamics simulation. Compared to the pristine CNCs, the mechanical properties of the CNCs-graphene layered structure can be prominently improved due to the covalent bond interactions and van der Waals forces at the interfaces. The elastic modulus negatively correlates with temperature but has a positive relationship with strain rates and content of the covalent bonds. The atomic structures of pristine CNCs and CNCs-graphene layered structure during the tensile process are analyzed to reveal the relationship between structure and mechanical behavior. These results have a particular incentive effect on theoretical research into polymers with layered structures and their practical applications. 

## Figures and Tables

**Figure 1 nanomaterials-12-04170-f001:**
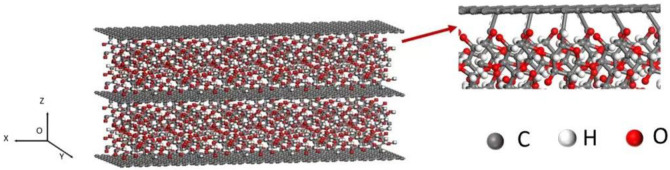
A schematic representation of the MD simulation model of CNCs-graphene layered structure.

**Figure 2 nanomaterials-12-04170-f002:**
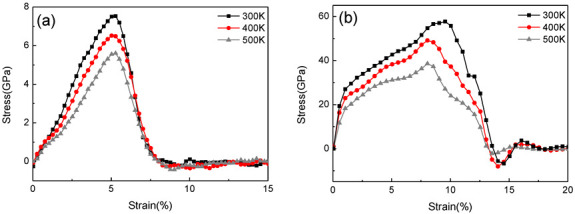
Stress-strain relationship at different temperatures. (**a**) Pristine CNCs. (**b**) CNCs-graphene layered composite.

**Figure 3 nanomaterials-12-04170-f003:**
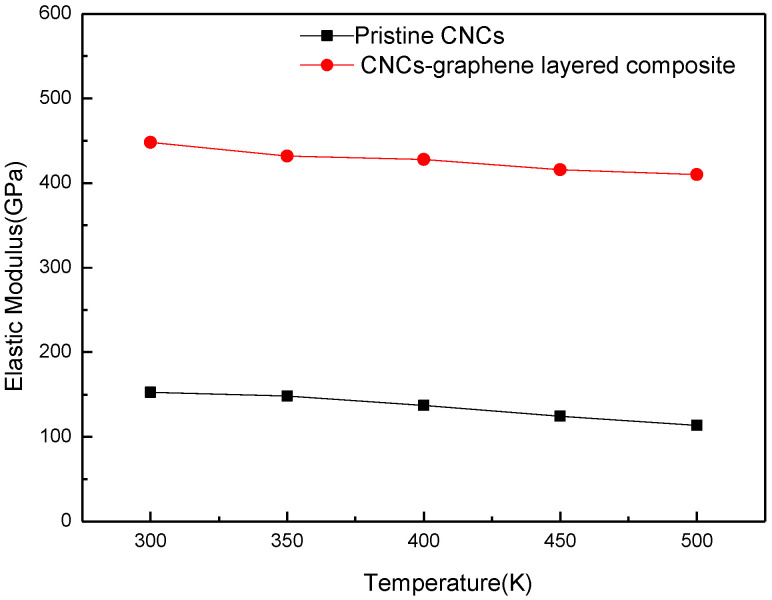
Comparison of elastic modulus of pristine CNCs and CNCs-graphene layered composite under different temperatures.

**Figure 4 nanomaterials-12-04170-f004:**
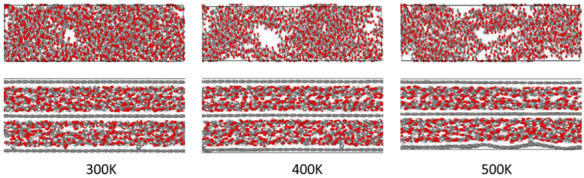
Visual representation of tensile process with the increasing temperature (upper atomic model is pristine CNCs and the lower atomic model is CNCs-graphene layered composite).

**Figure 5 nanomaterials-12-04170-f005:**
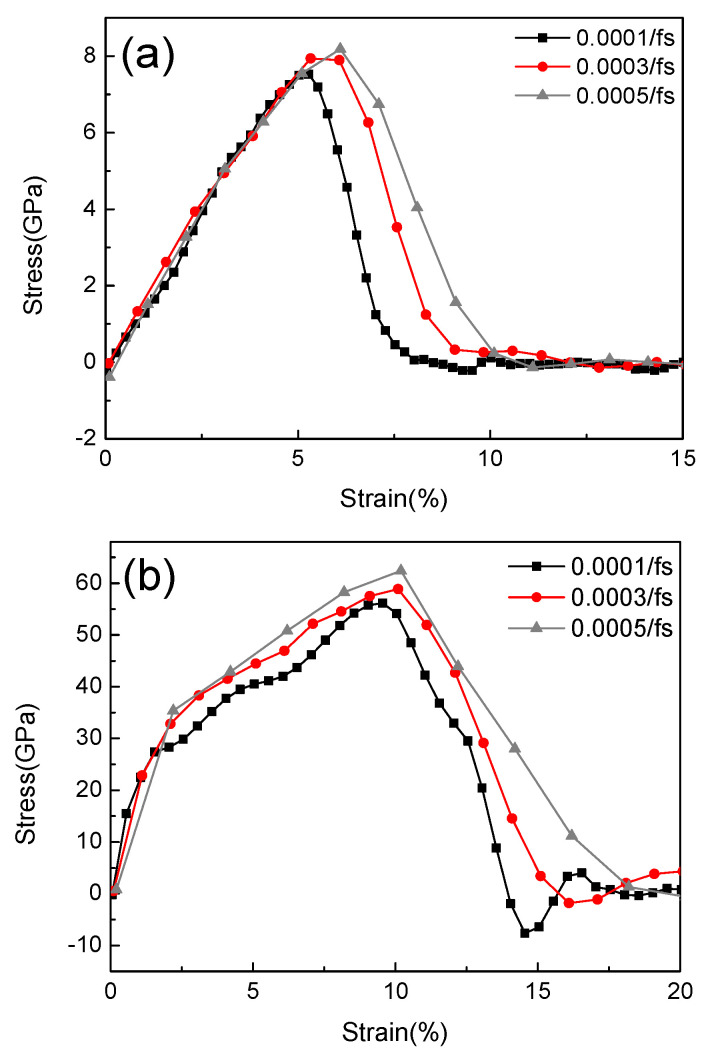
Stress–strain relationship at different strain rates. (**a**) Pristine CNCs. (**b**) CNCs-graphene layered composite.

**Figure 6 nanomaterials-12-04170-f006:**
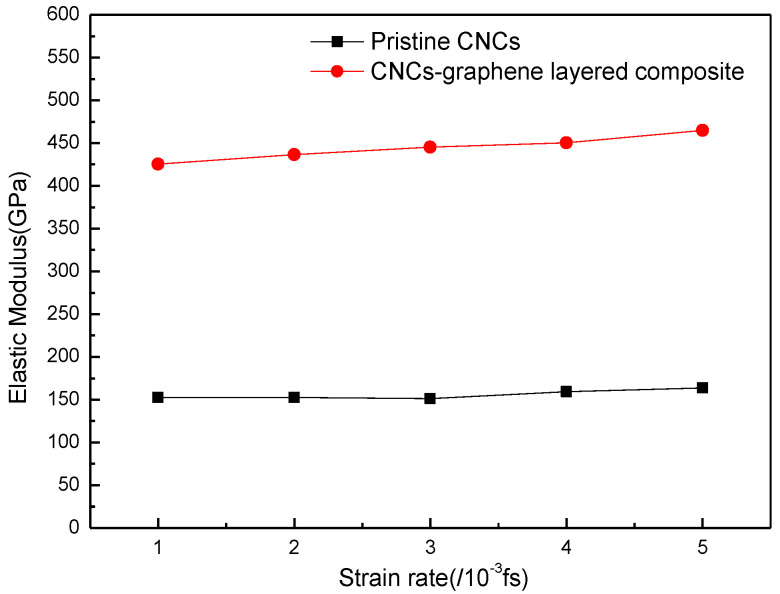
Comparison of the elastic modulus of pristine CNCs and CNCs-graphene layered structure under different strain rates.

**Figure 7 nanomaterials-12-04170-f007:**
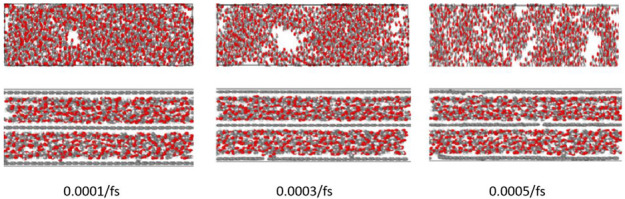
Visual representation of tensile process with different strain rates (Upper atomic model is pristine CNCs and the lower atomic model is CNCs-graphene layered composite).

**Figure 8 nanomaterials-12-04170-f008:**
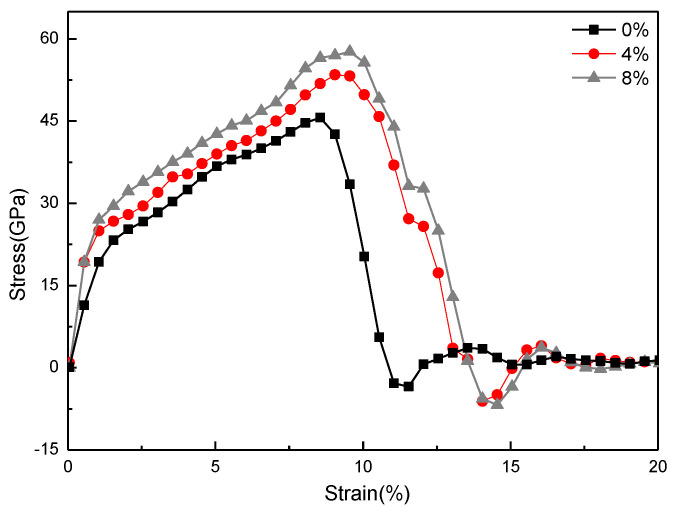
Stress-strain curve of CNCs-graphene layered structure with different covalent bond contents.

**Figure 9 nanomaterials-12-04170-f009:**
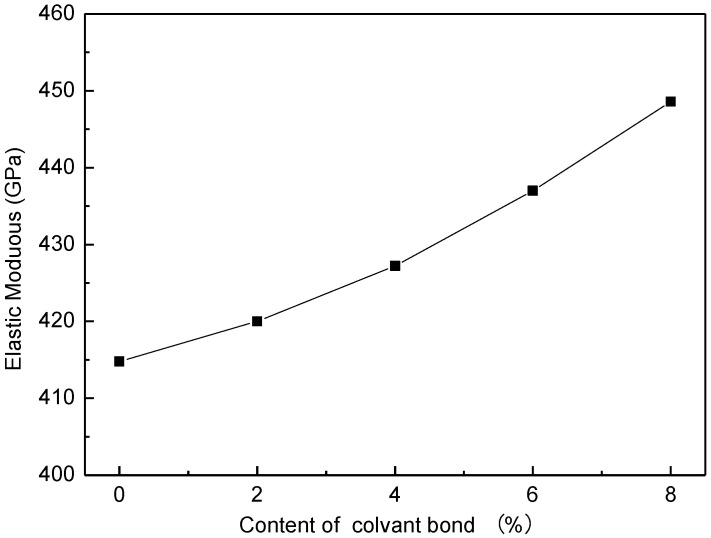
Elastic modulus of CNCs-graphene layered structure with different covalent bond content.

## Data Availability

The data presented in this study are available on request from the corresponding author.

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
