# Peer review of "Molecular Dynamics Simulations of the Mechanical Properties of Cellulose Nanocrystals—Graphene Layered Nanocomposites"

_nanomaterials, 2022, doi:10.3390/nano12234170_

Round 1
Reviewer 1 Report
The authors have performed classical molecular dynamics simulations of tensile-strain experiments on a model of cellulose nanocrystal sandwiched between parallel graphene planes. They have obtained stress-strain curves of this composite material at different temperatures, for five values of the strain rate and different covalent bonding at the interfaces. Elastic data deduced from these curves are compared to those of pure cellulose (nanocrystal model without graphene).
The results are interesting and understandable. However, a few points are puzzling and require explanations. They are listed herebelow.
1) Please, give the dimensions of the supercell used for the pure cellulose nanocrystal. What is the distance between the graphene planes in the composite model? If Fig. 1 represents the supercell, then the interplane distance should be half the size of the box in the Z direction, namely 1.15 nm. However, looking at Fig. 5 at 500 K, the bottom graphene plane is somewhat deformed, while the top plane is not. It means that the top graphene plane is not a periodic replica of the bottom graphene plane. One is then forced to conclude that Fig. 1 is not the full supercell.
2) The Berendsen thermostat has a bad reputation in molecular dynamics, as it may lead to artifacts, see doi:10.1021/acs.jctc.8b00446. Did the authors pay attention to that possibility and how? This is a particularly important issue when looking at temperature effects such as in Fig. 3.
3) Do the panels of Fig. 4 represent snapshots at a given strain for each temperature? Then, what strain value is illustrated?
4) Page 5: the slope at the origin of the black, red and gray curves in Fig. 5(b) obviously decreases, indicating a decrease of Young modulus with increasing strain rate. This behaviour is in contradiction with the red curve of Fig. 6 and with the sentence "the elastic modulus of CNCs-graphene layered composite increases with the increasing strain rates". Perhaps the way the authors evaluate the elastic modulus is responsible for this apparent discrepancy. In any case, explanations are required.
5) In the tensile simulations, did the authors allow the parameters of the supercell to relax in the y and z directions to accomodate the extension along the x direction?
6) Do the authors know how it would be possible to control the percentage of the cellulose covalent bonds to the graphene planes ?
A few minor additional points:
*) Page 2, last paragraph before section 2: something is wrong with the sentence "(NEMD) under typical temperature and strain rate. And predict the effect of covalent"
*) Very last word at the bottom of Page 2: "trails" or "trials?
*) Page 3, last sentence right above Fig. 2: "It can be proven that the simulation models constructed in this paper are valid." The author probably mean that the good agreement of their simulations with ref 21 indicate that the simulation models constructed in this paper are valid.
*) Fig. 5, please give the temperature value for the simulations with different strain rates.
Author Response
Thank you for your comments concerning our manuscript entitled “Molecular dynamics simulations of the mechanical properties of cellulose nanocrystals -graphene layered nanocomposites ”. Those comments are all valuable and very helpful for revising and improving our paper, as well as the important guiding significance to our researches. We have studied comments carefully and have made correction which we hope meet with approval.
(1)Please, give the dimensions of the supercell used for the pure cellulose nanocrystal. What is the distance between the graphene planes in the composite model? If Fig. 1 represents the supercell, then the interplane distance should be half the size of the box in the Z direction, namely 1.15 nm. However, looking at Fig. 5 at 500 K, the bottom graphene plane is somewhat deformed, while the top plane is not. It means that the top graphene plane is not a periodic replica of the bottom graphene plane. One is then forced to conclude that Fig. 1 is not the full supercell.
Response: Many thanks for your suggestion. According to the Iβ-type unit cell from the research of Nishiyama et al. [Journal of the American Chemical Society, 2002, 124(31): 9074-9082. ], the details atomic structure of CNCs is shown as below. The lattice parameters are a=7.784Å, b=8.201Å, c=10.38Å,α=β=90°,γ=96.5. The expanded lattice length of the CNCs structure used in this paper has a=2.3352nm, b=3.2804nm and c=6.228nm, which are basically consistent with the composite structure. The distance between the graphene layers is 1.56nm. We make a careful check of our paper, and find a mistake. The Y direction is actually 2.3nm, and the Z direction is 3.8nm. We have corrected them. Three graphene layer are directly constructed, and the top of graphene plane is not a replica of the bottom graphene plane.
(2)The Berendsen thermostat has a bad reputation in molecular dynamics, as it may lead to artifacts, see doi:10.1021/acs.jctc.8b00446. Did the authors pay attention to that possibility and how? This is a particularly important issue when looking at temperature effects such as in Fig. 3.
Response: We are very sorry for our incorrect writing. The Nosé-Hoover thermostat is actually adopted in this paper which could avoids the shortcomings of Berendsen thermostat. The Nosé-Hoover thermostat can adjust the temperature of the system by changing the Hamiltonian of the system and exchanges energy.
(3)Do the panels of Fig. 4 represent snapshots at a given strain for each temperature? Then, what strain value is illustrated?
Response: The changes of atomic structure of pristine CNCs and CNCs-graphene layered composites under different temperature are shown in Fig. 4. The elastic deformation become more obvious as the temperature increases, which could verify that the elastic modulus of all models decrease with increasing temperature. The CNCs chains in CNCs-graphene layered structure have no apparent breakage, which could indicate that the elastic modulus of it is much higher than that of pristine CNCs.
(4)Page 5: the slope at the origin of the black, red and gray curves in Fig. 5(b) obviously decreases, indicating a decrease of Young modulus with increasing strain rate. This behaviour is in contradiction with the red curve of Fig. 6 and with the sentence "the elastic modulus of CNCs-graphene layered composite increases with the increasing strain rates". Perhaps the way the authors evaluate the elastic modulus is responsible for this apparent discrepancy. In any case, explanations are required.
Response: The ReaxFF force field used in this paper could not match the composite structure accurately, and there is no specific potential function for that. The local singularity phenomenon appear in the simulation. Thus, all the values of the liner region are considered as in the calculation of Young’s modulus for eliminating the singularity phenomenon in that local region as much as possible. The same method was used in the following reference. [Nanomaterials, 2020, 10(1): 154. DOI: 10.3390/nano10010154]
(5)In the tensile simulations, did the authors allow the parameters of the supercell to relax in the y and z directions to accomodate the extension along the x direction?
Response: Many thanks for your comments. The parameters of the supercell to relax in the y and z directions are allowed to accomodate the extension along the x direction.
(6)Do the authors know how it would be possible to control the percentage of the cellulose covalent bonds to the graphene planes ?
Response: The cellulose covalent bonds are generated by randomly removing H atoms in CNCs chains and then the bonds were bonded to a carbon atom of the graphene layer. Thus, the coverage of the covalent linkages is defined as the number of the covalent linkages of each layer divided by the total number of carbon atoms of grapheme layer.
A few minor additional points:
*) Page 2, last paragraph before section 2: something is wrong with the sentence "(NEMD) under typical temperature and strain rate. And predict the effect of covalent"
Response: We are very sorry for our incorrect writing, and the grammar mistakes have been revised.
*) Very last word at the bottom of Page 2: "trails" or "trials?
Response: Many thanks for your comments. The word has been modified.
*) Page 3, last sentence right above Fig. 2: "It can be proven that the simulation models constructed in this paper are valid." The author probably mean that the good agreement of their simulations with ref 21 indicate that the simulation models constructed in this paper are valid.
Response: Many thanks for your comments. The word has been modified.
*) Fig. 5, please give the temperature value for the simulations with different strain rates.
Response: Many thanks for your comments. We have added the temperature condition.

Reviewer 2 Report
Comments from Reviewer
Title: Molecular dynamics simulations of the mechanical properties of cellulose nanocrystals -graphene layered nanocomposites
The current form's presentation of methods and scientific results is satisfactory for publication in the Nanomaterials journal. The minor and significant drawbacks to be addressed can be specified as follows:
1. Page 1. [12-13] ---> [12, 13]. See Page 2, e.g. [29,30].
2. Page 2, LAMMPS package. No reference(s).
3. Fig. 4, 500K. The bottom layer of graphene is deformed. Why only this one?
Sincerely,
The reviewer.
Author Response
Thank you for your comments concerning our manuscript entitled “Molecular dynamics simulations of the mechanical properties of cellulose nanocrystals -graphene layered nanocomposites ” . Those comments are all valuable and very helpful for revising and improving our paper, as well as the important guiding significance to our researches. We have studied comments carefully and have made correction which we hope meet with approval.
(1)Page 1. [12-13] ---> [12, 13]. See Page 2, e.g. [29,30].
Response: Many thanks for your comments. The mistake has been modified.
(2)Page 2, LAMMPS package. No reference(s).
Response: We have added the citation for the LAMMPS package in references.
(3) 500K. The bottom layer of graphene is deformed. Why only this one?
Response: Many thanks for your comments. The covalent bondings between CNCs and graphene layers start to break due to the increasing temperature. The broken covalent bonding results in high tensile strength along with graphene layer. Thus, the bottom layer of graphene become crinkle.
Reviewer 3 Report
The authors report competent work on the NEMD simulation of CNC-graphene layered structures based on a 6.4 nm x 3.8 nm x 2.3 nm assembly. They exploit the agreement between their results and the experimental mechanical data known for pristine CNC from reference [21] as a justification for their modelling techniques when extended to CNC-graphene layered composites.
An increase in the elastic modulus by a factor of approximately three is reported when going to the composites. They conclude further that temperature has a deleterious effect on the elastic modulus of the composites, whereas strain rate behaviour is improved. They report that covalent bond coverage also has a positive effect on the elastic modulus.
The work presented cannot be subjected to scientific scrutiny, since it is simply a report of the results obtained by their modelling work. The results are plausible, however.
I recommend publication, since the paper is succinctly written and will be of interest to workers in the limited field of cellulose nanocrystals.
Author Response
Many thanks for your comments.
Round 2
Reviewer 1 Report
I am grateful to the authors for their comments. However, I am not totally satisfied, as the authors do not really answer the questions of Reviewers 1 and 2.
My question 1 contained question 3 of Reviewer 2: why do the bottom graphene layer deform and only that one. I do not think the authors have answered that point.
I do not understand the answer to questions 3 and 4. Please reformulate. Question 6 was about experiment, not simulation. If one had to prepare a multilayered structure of the kind discussed here, could it be possible to control the percentage of covalent bonds to the graphene sheets?
Author Response
We are truly grateful to reviewer’s critical comments and thoughtful suggestions which would help us in depth to improve the quality of the paper.In the following you will find our point-by-point responses to the reviewers’ questions.
- Why do the bottom graphene layer deform and only that one. I do not think the authors have answered that point.
Response: Many thanks for your comments. The covalent bondings between CNCs and graphene layers play an important role in exterior force resistance. Before the covalent bondings destroyed, the hydrogen bonds and intra-chain hydrogen bonds in CNCs are not destroyed due to the increasing temperature. The covalent bondings break and generate a high tensile force along the graphene plane, which result in a deform of the graphene layer. A similar phenomenon can be found in the research of Shishehbor et al [Nanomaterials, 2020, 10(1): 154. DOI: 10.3390/nano10010154]. It can be seen in the figures, the carbon nanotubes applied different potential functions also break and deform starting from the bottom.
- I do not understand the answer to questions 3 and 4. Please reformulate.
Do the panels of Fig. 4 represent snapshots at a given strain for each temperature? Then, what strain value is illustrated?
Response: Many thanks for your comments. The atoms configuration of two nanocomposites under tensile stresses with the increasing temperatur could be only verify the variation trend of strain, and can’t give the strain value. As shown in Fig. 4, the elastic deformation become more obvious as the temperature increases, which could verify that the elastic modulus of all models decrease with increasing temperature.
the slope at the origin of the black, red and gray curves in Fig. 5(b) obviously decreases, indicating a decrease of Young modulus with increasing strain rate. This behaviour is in contradiction with the red curve of Fig. 6 and with the sentence "the elastic modulus of CNCs-graphene layered composite increases with the increasing strain rates". Perhaps the way the authors evaluate the elastic modulus is responsible for this apparent discrepancy. In any case, explanations are required.
Response: Many thanks for your comments. The local singularity phenomenon appear in the simulation. All the values of the liner region are considered as in the calculation of Young’s modulus. A similar research occurred in the research of Wu et al [ Cellulose,2014,21(4):2233-2245]. As seen in the following figures, some calculation values in molecular dynamics simulation are not accurate enough. In order to better study its overall change trend of Young's modulus, the strain range of 0-10% is selected as simulation range in this paper.
- Question 6 was about experiment, not simulation. If one had to prepare a multilayered structure of the kind discussed here, could it be possible to control the percentage of covalent bonds to the graphene sheets?
Response: Many thanks for your comments. We have prepare the CNCs-graphene layered structure by vacuum filtration method. The percentage of covalent bonds are depended on the concentration of graphene suspension and reaction time. So we get different percentage of covalent bonds by controlling the the concentration of graphene suspension and reaction time.
